# RoboAgent:
# Generalization and Efficiency in Robot Manipulation via Semantic Augmentations and Action Chunking

**Homanga Bharadhwaj**[*]        **Jay Vakil**[*]        **Mohit Sharma**[*]

**Abhinav Gupta**        **Shubham Tulsiani**        **Vikash Kumar**

Carnegie Mellon University        FAIR, AI at Meta

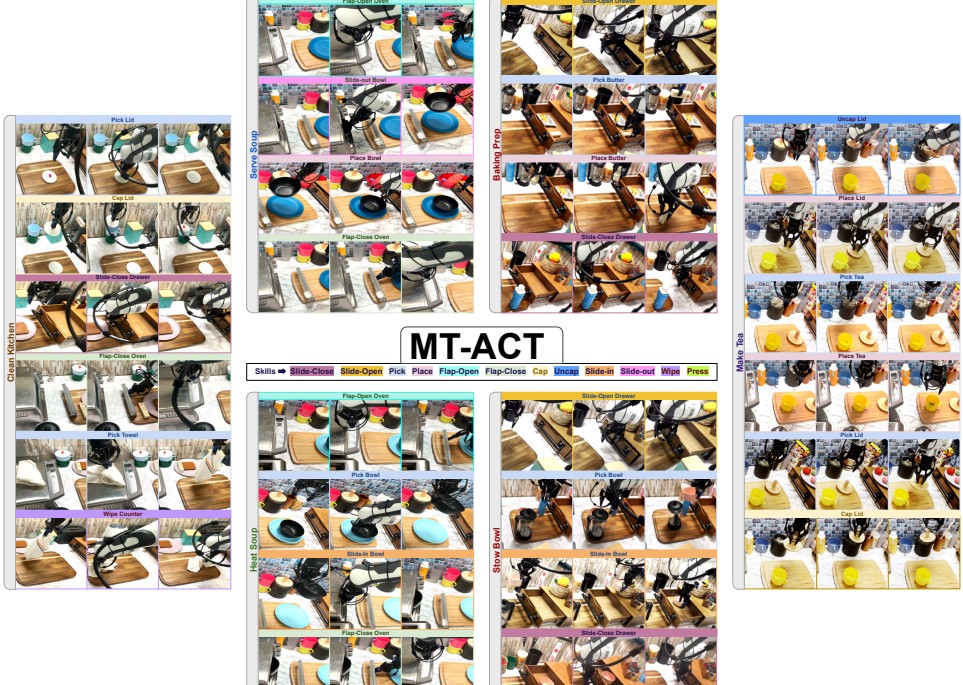

Figure 1: A glimpse of the diverse manipulation capabilities of *RoboAgent*– a single agent trained with a novel MT-ACT architecture capable of 12 manipulation skills across 38 tasks encompassing 6 activities. https://roboagent-anonymous.github.io/

**Abstract:** The grand aim of having a single robot that can manipulate arbitrary objects in diverse settings is at odds with the paucity of robotics datasets. Acquiring and growing such datasets is strenuous due to manual efforts, operational costs, and safety challenges. A path toward such a universal agent requires an efficient framework capable of generalization but within a reasonable data budget. In this paper, we develop an efficient framework (MT-ACT) for training universal agents capable of multi-task manipulation skills using (a) *semantic augmentations* that can rapidly multiply existing datasets and (b) *action representations* that can extract performant policies with small yet diverse multi-modal datasets without overfitting. In addition, reliable task conditioning and an expressive policy architecture enables our agent to exhibit a diverse repertoire of skills in novel situations

7th Conference on Robot Learning (CoRL 2023), Atlanta, USA.

specified using task commands. Using merely 7500 demonstrations, we are able to train a single policy *RoboAgent* capable of 12 unique skills, and demonstrate its generalization over 38 tasks spread across common daily activities in diverse kitchen scenes. On average, *RoboAgent* outperforms prior methods by over 40% in unseen situations while being more sample efficient. Videos and details are in the website https://roboagent-anonymous.github.io/

# 1 Introduction

Developing a robot manipulator with multiple skills requires exposure to diverse experiences and the ability to acquire skills from a diverse data corpus. Collecting such multi-skill data corpus in the real world requires substantial effort and suffers from high operational costs and safety challenges. Given the expense, efficiency in robot learning paradigms is necessary for real-world training and deployment. While there are recent efforts in scaling real-world robotic datasets despite these challenges [1, 2, 3], efficiency seems to be overlooked in the attempts to scale [4, 5, 6, 7].

With the acknowledgment that robot learning will generally benefit as the scale of the robotics dataset grows, the focus of our work is on investigating generalization in developing capable agents under a *given data budget*. We restrict ourselves to a dataset with 7,500 robot manipulation trajectories (an order of magnitude less than related works [5]) containing a diverse collection of manipulation skills across different tasks. As a robot under deployment in real environments like homes, hospitals, etc., will always find itself in unseen scenarios, we set out to develop the most capable agent with an emphasis on its *ability to generalize to novel situations within this data budget*.

At first sight, wide generalization with a data budget seems like wishful thinking - while it's possible to provide large representation capabilities to the agent's policy, scaling without data diversity will likely lead to overfitting and no generalization. Our insight is twofold: (1) collect a reasonably sized dataset (7,500 trajectories) with diverse coverage of skills, and devise a semantic augmentation strategy to rapidly multiply the dataset without additional human / robot cost, (2) devise a language-conditioned multi-task multi-scene policy architecture capable of handling the multi-modal data distribution. The architecture leverages the fact that robot movements are temporally correlated, by predicting action chunks [8] instead of per-step actions, leading to smoother behaviors and mitigation of covariate shift commonly observed in the low data imitation learning regime.

Overall, we emphasize that the data efficiency lessons we present are *general* and will help in achieving generalizable agents independent of the available data budget. Building on these insights, we make the following contributions:

- We present an efficient method MT-ACT designed to recover **generalist agents on a data budget**. MT-ACT leverages data multiplication via semantic augmentations and action representations to drive efficiency gains in low-data settings.
- MT-ACT's architecture can effectively ingest multi-modal trajectory data to recover *RoboAgent* – a single policy that can perform a diverse set of tasks through language instructions. Through extensive real-world experiments, we show *RoboAgent* is **capable of exhibiting 12 manipulation skills**.
- We perform extensive generalization studies to demonstrate that MT-ACT is 40 % more performant than alternatives, exhibits **superior generalization to diverse novel scenarios**, is amenable to **improvements and extensions during deployment through fine-tuning** and is robust for reproduction in completely new geographical setups.
- We meticulously recorded all the data collected during the course of the project which we are open-sourcing as part of `RoboSet` - one of the **largest open-source robotics dataset** on commodity hardware. It contains high-quality human teleOp trajectories spanning a balanced distribution of 12 skills across 38 tasks in diverse kitchen scenes.

# 2 MT-ACT: Multi-Task Action Chunking Transformer

To learn generalizable manipulation policies, robots require rich and diverse experiences, encompassing a wide range of skills and contextual variations. However, operational costs and real-world

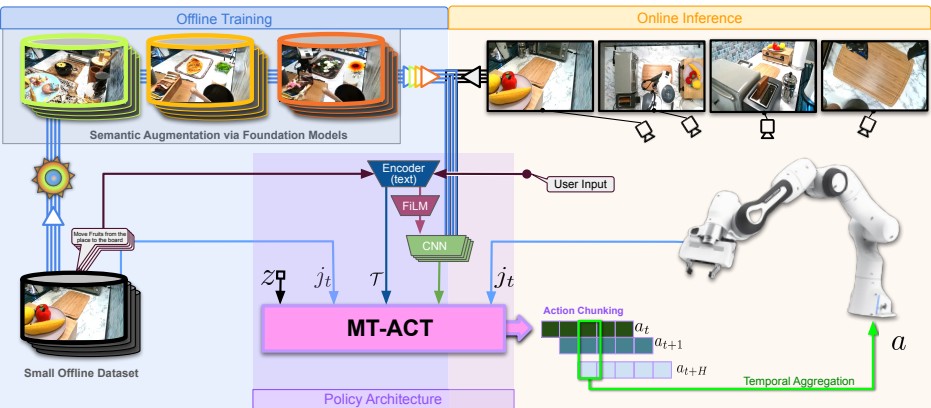

Figure 2: Two stage framework: [Left] **Semantic augmentation** stage diversifies the robot data offline using inpainting augmentations at no extra human/robot cost. [Right] **Policy learning** stage trains language-conditioned policy using MT-ACT – multi-task action-chunking transformers – which leverages efficient action representations for ingesting multi-modal multi-task data into a single multi-skill multi-task policy.

challenges in collecting such extensive datasets pose a practical limit on their overall size. We address these limitations by developing *a paradigm that can learn effective multi-task agents under a limited data budget*. Our approach consists of two stages (Figure 2):

**Semantic Augmentation** – the first stage multiplies the pre-collected dataset by creating a diverse collection of semantic augmentations over existing robot's experiences. These semantic augmentations recreate a single robot demonstration into several demonstrations, each with a different semantic context (objects, textures, backgrounds, etc), at no extra robot or human cost. Such data diversification incorporates real-world semantic priors to make the multi-task agent robust to test-time out-of-distribution scenarios.

**Policy Learning** – the second stage learns robust skills from limited skill data by adapting design choices from prior works limited to single-task settings for large-scale generalization in multi-task multi-scene manipulation tasks. We develop MT-ACT – a language-conditioned novel policy architecture to train robust agents capable of recovering multiple skills from multi-modal datasets.

## 3 Experiments

Our experiments aim to understand the following questions

- How does MT-ACT perform, quantitatively and qualitatively, on a large set of vision-based robotic manipulation tasks? How does it generalize to new tasks, objects, and environments?
- Does semantic augmentation improve policy generalization (i.e. scenes with new target objects)?
- Does action chunking help with temporally consistent trajectories, achieving higher success?

To answer these research questions we instantiate our framework in the real world using commodity hardware and objects commonly used in everyday kitchens. We provide details of data, setup, and baselines, and explanation of results in the Appendix.

### 3.1 Multi-Task Real-World Results

**Performance.** Figure 3 (Right) compares our proposed MT-ACT policies against commonly used imitation learning architectures. We show success rates on the y-axis, with 20 evaluation rollouts per task, averaged over all tasks. In this figure (Figure 3 Left-Bottom) we only plot results for *L1-generalization* since this is the standard setting most other imitation learning algorithms use. We observe that all approaches that only model next-step actions (instead of sub-trajectories) exhibit weaker performance. Among these approaches, we find that action-clustering-based approaches

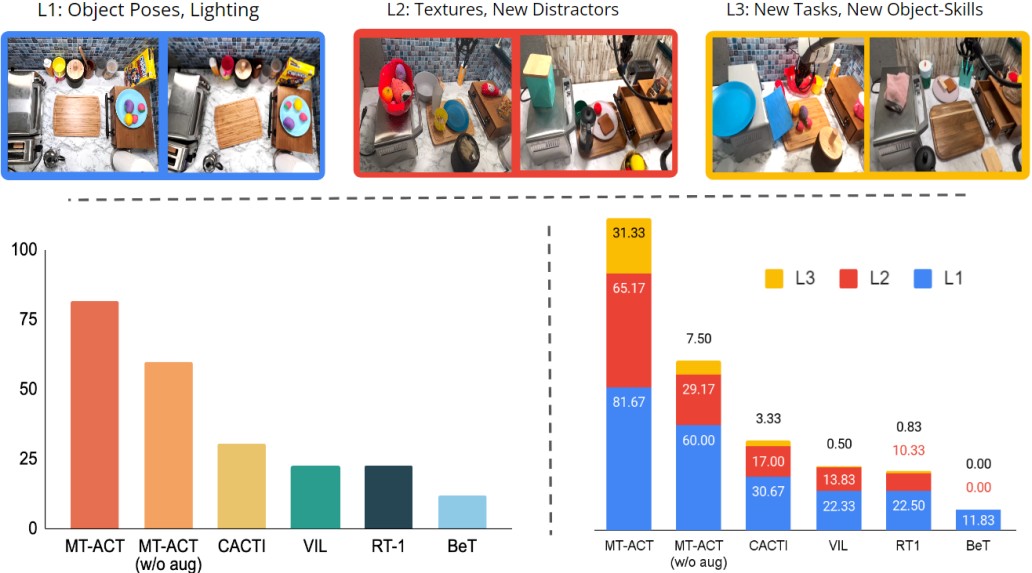

Figure 3: Visualization of different generalization axes, evaluating effectiveness with lighting variations and smaller scene changes such as object poses (L1), robustness to significant scene variations (L2), generalization to unseen tasks (L3). *Bottom-Left:* Results for commonly used L1-generalization. *Bottom-Right:* Multi-Task (universal policy) results for different levels of generalization. See 9 for L4-generalization results.

(BeT [9]) for multi-task settings, perform significantly worse. We believe this happens because naive clustering in very diverse action distributions may not result in clusters that generalize across diverse skills. Additionally, since we are operating under a data budget, we observe that RT1-like approaches that require a lot of data do not perform well in the low data regime. By contrast, our MT-ACT policy which uses action-chunking and CVAE to model multi-modal sub-trajectories significantly outperforms all baselines.

**Generalization and Robustness.** Figure 3 (Bottom-Right) shows the results for all methods across multiple levels of generalization (**L1**, **L2**, and **L3**). Recall that these levels of generalization include diverse table backgrounds, distractors (**L2**) and novel skill-object combinations (**L3**). From Figure 3 (Bottom-Right) we see that by virtue of semantic augmentations and action representations, MT-ACT significantly outperforms all the baselines we consider. More interestingly, we see that semantic augmentations have less effect for L1-generalization ($\approx 30\%$ relative), they provide a *much more* significant improvement for both L2-generalization ($\approx 100\%$ relative) and L3-generalization ($\approx 400\%$ relative). Since semantic augmentations affect both scenes (backgrounds and distractor objects) as well as target objects being manipulated they provide useful support for the policy to achieve increasing levels of generalization.

## 4 Discussion

We develop a framework for sample-efficient and generalizable multi-task robot manipulation in the real world. Our framework is based on rapidly multiplying a small robotics dataset through semantic augmentations, and training a language-conditioned policy that can ingest the diverse multi-modal data. We combine and adapt several design choices like action chunking and temporal aggregation proposed in the context of single-task policies, and show that they yield significant boosts in performance even in multi-task settings. We also release one of the largest manipulation datasets to date involving over 12 skills in kitchen environments which we hope will facilitate further research in developing diverse real-world robot manipulation systems. A limitation of our work is that we do not consider composing skills across similar/different activities. Another limitation is that we do not explore the axes of language generalization, and use language embeddings from pre-trained encoders as is. Future work could investigate better language conditioning that is more flexibly adaptable to changes in task descriptions.

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

# Appendix

## A  Related Work

**Frameworks for Scaling Robot Learning.** Given the cost of supervision in robot learning, self-supervised learning [10, 11, 12] methods leveraging large unlabeled datasets have been a dominant paradigm towards building general-purpose agents. Large-scale simulations [13, 14, 15, 16] have also been leveraged with the hope of first learning a general multi-task policy [17, 18, 19, 20, 21, 22] and then transferring it to real world via sim2real[23, 24, 25, 26]. However, most multi-task RL works focus on narrow domains [20, 27], and those in the real-world show limited generalization and task diversity [28, 29]. While other works [17, 18, 30] focus on diverse multi-task scenarios, they restrict to evaluating trained policies mostly in simulation. By contrast, our work focuses on a large, diverse set of real-world manipulation tasks. Many recent works use imitation learning with large-scale real-world robot tele-operation datasets [2, 3, 31, 1, 32, 33] . While early works collect limited real-world data [3, 32], more recent approaches [1, 5, 7] collect much larger datasets. In fact, [5] gathers, possibly, the largest dataset ($\approx 130K$ demonstrations) outside bin and place settings and shows impressive generalization with skills learned using this data. Our work is similar in spirit, *i.e.*, we focus on real-world manipulation tasks and aim to learn a multi-task policy using *limited* real-world demonstrations. However, unlike [1], we avoid toy environment setups and focus on realistic real-world kitchen setups with clutter and multiple feasible tasks in a scene. Additionally, our agents exhibit a much greater diversity of skills than [5, 7, 26] while being trained only on 7.5k trajectories, as opposed to 135k in [5]. Importantly, we collect our data using commodity hardware (see Figure 7) and make it readily available to robotic researchers worldwide.

**Alternate Data Sources in Robotics.** Recent successes of large-scale self-supervised approaches within both language and vision communities have showcased the advantage of large-scale data. Many recent works propose using pre-trained visual representations trained primarily on non-robot datasets [34, 35], for learning control policies [36, 37, 24, 38, 39]. Most of these works focus on single-task settings [36, 37, 40, 41], or simulated robot environments [41, 38]. Given challenges with collecting *large* real-world robotics datasets, some works focus on alternate data sources such as language [42, 43, 44, 45], human videos [46, 47, 48, 49, 50, 51, 52, 53], and generative augmentations [54, 55, 29, 56, 57]. Our work is similar to the latter set of works, some of which use diffusion models to generate augmentations for data collected in the real world. However, unlike [29, 56] our approach is fully automatic. We do not need segmentation masks [29] or object meshes [56] for generating augmentation data. Our work is most similar to [57] which adapts a pre-trained open-world object detection model [58] for generating segmentations that are used with text-guided diffusion models to generate augmentations. However, our approach does not require any further fine-tuning of a separate module for open-vocabulary segmentation and language grounding. We further investigate scaling semantic data augmentations and demonstrate its favorable impact on test-time generalization.

## B  RoboAgent Framework Details

### B.1  Dataset (`RoboSet`)

Training a general agent capable of robustly exhibiting a diverse repertoire of skills in novel scenes and tasks needs exposure to experiences matching this diversity. To align with our goal of building a data-efficient robot learning paradigm, we restrict ourselves to a frozen pre-collected small but diverse dataset – `RoboSet`(MT-ACT). To capture behavioral diversity, we ensure sufficient coverage over different core skills, where each skill if defined as a temporally correlated sequence of actions that lead to plausible change in an object's pose. Example skills include *closing/opening* articulated objects, *sliding*, *wiping*. Each skill is instantiated across a set of objects. We refer to such (skill, object) combinations as a **task**. Our tasks are instantiated in different kitchen scenes, visually illustrated in Appendix (see webpage). Instead of a random collection of tasks, we structure groups of

|  | **Trajectories** | **Tasks** | **Skills** | **Scenes** | **Source** |
|---|---|---|---|---|---|
| RoboSet (**MT-ACT**) | 7,500 | 38 | 12 | 10 | TeleOp |
| RoboSet (**kitchen**) | 30,050 | 38 | 12 | 10 | TeleOp |
| RoboSet (**bin**) | 70,000 | 10 | 4 | 1 | Heuristics |
| RoboSet (**full**) | 98,050 | 48 | 12 | 11 | TeleOp+Heuristics |
| BridgeData [1] | 33,200 | 72 | 8 | 10 | TeleOp |
| BC-Z [6] | 25,000 | 100 | 9 | N/A | TeleOp |
| RoboTurk [3] | 2,100 | N/A | 3 | 1 | TeleOp |
| Amazon Pick-Place [59] | 100,000 | N/A | 1 | 1 | Heuristics |
| RoboNet [2] | 162,000 | N/A | 2 | 7 | Heuristics |
| BAIR Pushing [60] | N/A | N/A | 1 | 1 | Heuristics |

Table 1: Open-source real-world manipulation dataset landscape: RoboSet(ours) https://sites.google.com/view/mtact/roboset is one of the largest open-source robotics datasets. It contains high-quality demonstration, including human tele-operation, trajectories spanning a balanced distribution of 12 skills across 38 tasks in diverse kitchen scenes.

tasks as belonging to be part of a household **activity**, such that they can be executed in sequence to obtain a meaningful outcome, such as cleaning a kitchen.

RoboSet(MT-ACT) – the dataset we used for this project (i.e. to train *RoboAgent*) consists of 7,500 trajectories (Table 1)[1] collected using human teleoperation. The dataset involves 12 skills (see Figure 6 for skill distribution). While the common pick-place skills cover 40% of the dataset, we also include contact-rich skills (*Wipe*, *Cap*) as well as skills involving articulated objects (*Flap-Open*, *Flap-Close*). We collect the overall dataset across four different physical setups. Each setup is instantiated with various everyday objects to create a kitchen scene. We frequently vary each set up with different variations of objects, thereby exposing each skill to multiple target objects and scene instantiations. Figure 7 provides a glimpse of the overall setup and a subset of objects. Overall, unlike previous datasets, RoboSet provides a broad coverage of manipulations skills for generalist robots required to operate in kitchen environments.

In Table 1, we compare our dataset with existing *open-source* robot manipulation datasets. As noted above, we use RoboSet(MT-ACT) (7.5K) trajectories to train *RoboAgent*. However, we release a much larger dataset, RoboSet which includes more teleoperated data, data collected during policy evaluation and data for non-kitchen settings. Overall, the entire RoboSet is one of the largest publically released datasets with commodity robots and collected in real-world setup. RoboSet contains a large number of diverse skills and scene variations.

## B.2 Semantic Data Augmentation

Generally useful robot manipulation systems will need to be able to deal with out-of-distribution scenarios (e.g. different homes and offices). Since any dataset of a practical size will have a limited diversity of objects and scenes (due to physical access and operational constraints) compared to what agents will encounter during deployment, we develop a *fully automatic* offline process to multiply our dataset.

Given an initial dataset of robot behaviors, we multiply the dataset by creating multiple semantic variations of the dataset while preserving the *robot behavior* within each trajectory. These semantic variations are created by applying augmentations per frame within the trajectory. Augmentations are created by inpainting a part of the image frame introducing new objects and scene variations. The inpainting locations are specified by a mask and are informed by a text prompt. As opposed to [29, 56, 57] needing manual masks, object templates, etc., our approach is fully automatic. We use the SegmentAnything model [61] to automatically detect semantic boundaries in the scene to create augmentation masks. See Section B.3 for additional details. We emphasize that our approach toward

---

[1]Note that the entire RoboSet is much larger and much more diverse. *RoboAgent* is trained on RoboSet(MT-ACT) – a subset consisting of 7500 trajectories

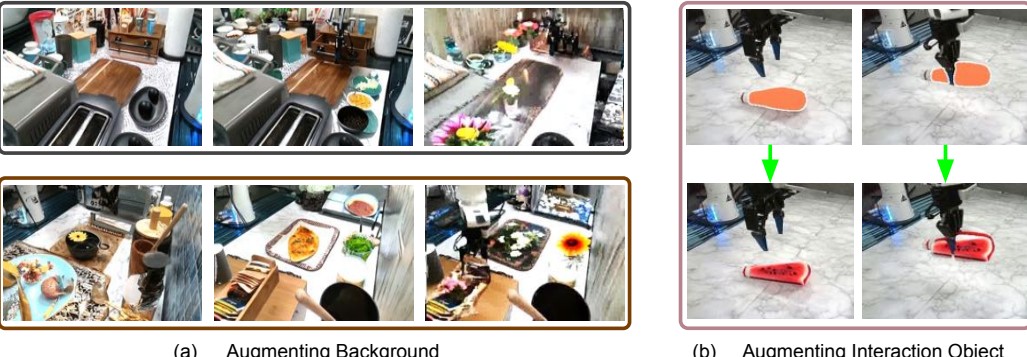

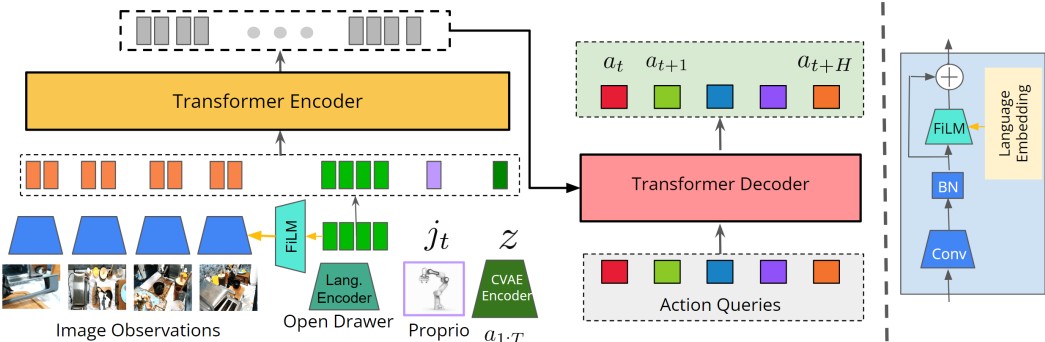

(a)   Augmenting Background            (b)   Augmenting Interaction Object

Figure 4: Illustration of the data augmentations used to rapidly multiply limited robot datasets with diverse semantic scene variations. (a) shows the scene around the robot and the interaction object changing. (b) shows the interaction object itself changing while preserving the rest of the scene.

Figure 5: Policy architecture for MT-ACT . We use a CVAE that learns latent encodings $z$ for action sequences to implicitly identify different *modes* in the data. A transformer takes as input a latent code, language embedding of the task, and image embeddings from four camera views, to autoregressively output an action sequence $a_{t:t+H}$ for chunk size $H$. On the right, we shows details for the FiLM layer [63] that we use for language-conditioning.

semantic augmentation is fully automatic and offline. It takes advantage of and is also well poised to continually benefit from rapidly advancing progress in segmentation and in-painting models [61, 62]. Akin to fields of natural language processing and computer vision, by distilling semantic real-world priors present in internet images/videos into robotics datasets, it provides robot learning a scalable mechanism to benefit from internet-scale data at no extra cost to humans/robots.

### B.3   MT-ACT Policy Learning

Recovery of a generalizable robot manipulation policy under a practical data budget available in robotics demands an efficient policy architecture. In scenarios that have sufficient coverage within the training data, we want the policy to stay close to nominal behaviors (efficient imitation). The policy also needs to be effective to new variations (effective generalization) and contexts (efficient task conditioning) that are unseen during training . In addition, we want the policies to exhibit temporally correlated smooth behaviors accomplishing tasks with minimal errors and safety violations.

Our policy architecture – MT-ACT is designed to be a Transformer model [64] of sufficient capacity that can handle multi-modal multi-task robot datasets. In order to capture multi-modal data, following prior works [8] we incorporate a CVAE [65] that encodes action sequences into latent *style* embeddings $z$. The decoder of the CVAE is the Transformer policy that conditions on the latents $z$. This formulation of expressing the policy as a generative model helps in effectively fitting to the multi-modal teleop data, without ignoring regions of a trajectory crucial for precision, which are also likely to be more stochastic. In order to model multi-task data, we incorporate a pre-trained language encoder [66] that learns an embedding $\mathcal{T}$ of a particular task description. To mitigate issues of compounding error and to achieve smooth temporally correlated robot motions, at each time-step, we

predict actions $H$ steps in the future and execute them through temporal-aggregation of overlapping actions predicted for a particular time-step [8]. To improve effectiveness towards scene variations and robustness towards occlusions in clutter, we provide the policy with four different views of the workspace through four cameras.

At time-step $t$, the transformer encoder takes four camera views , $o_t^{1:4}$, the joint pose of the robot $j_t$, the style embedding from the CVAE $z$, and the language embedding $\mathcal{T}$. We use a FiLM-based conditioning [63, 5], in order to ensure that the image tokens are able to reliably focus on the language instruction, such that the policy doesn't get confused about the task when multiple tasks are possible in a scene. The encoded tokens go to the decoder of the Transformer policy with fixed position embeddings, which finally outputs the next action chunk ($H$ actions) for the current time-step. For execution, we average over all overlapping actions predicted for the current time-step (As $H > 1$, the action chunks overlap), and execute the resulting averaged action. Overall, our proposed architecture extends ACT [8] to multi-task ACT (MT-ACT) using appropriate language conditioning (see Section D.1). Since RoboSet(MT-ACT) contains diverse skills we show that the VAE prior can capture such behavior diversity. Finally, we demonstrate for the first time that action-chunking and temporal aggregation are useful for learning diverse multi-task behaviors for quasi-static (low-frequency control) tasks in diverse scenes.

## C  Experiment Details

**Robot hardware.** As noted before, Figure 7 shows our robot environment, called *RoboPen* that consists of a kitchen setup with everyday objects, a Franka Emika Panda arm with a two-finger gripper with adaptive fingers, three fixed cameras (top, left, right), and a wrist camera. We utilize all cameras for robust policy learning.

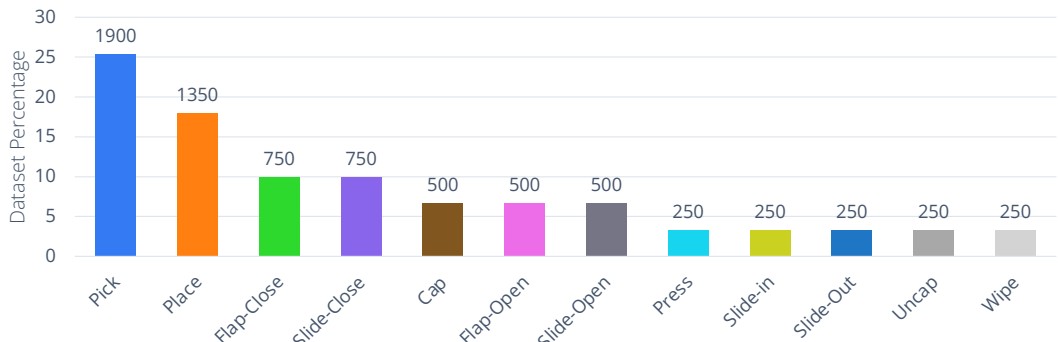

Figure 6: Skill distribution in terms of % of trajectories with a certain skill used to *train RoboAgent*. Number on top shows number of trajectories.

**Data collection.** As noted in (Section B.1) our tele-operated dataset is collected across four different physical setups with periodically changing kitchen-like environments. Additional details regarding the dataset, along with sample trajectories, and a link to the entire dataset are in the project website☐. We are publicly releasing this dataset, as part of RoboSet – a large multi-skill robotics dataset. To our knowledge, this is one of the largest open-source robot manipulation datasets with the most commonly used *non-proprietary robot hardware* (Franka Panda [67]) containing diverse real-world behaviors beyond pick and place.

**Generalization Axes.** Following prior work [5, 6, 18], we define each *task* to consist of a particular language command like *'pick a cube of butter from the drawer on the left'* that defines an object to be interacted with (butter), a skill to be executed (pick), and some context (drawer on the left). Each activity consists of a collection of 4-5 close tasks that can be executed in sequence. The policy trained to achieve (all tasks of) an activity is referred to as *activity policy* and the policy trained over all the activities as the *universal policy*. We consider different levels of generaliza-

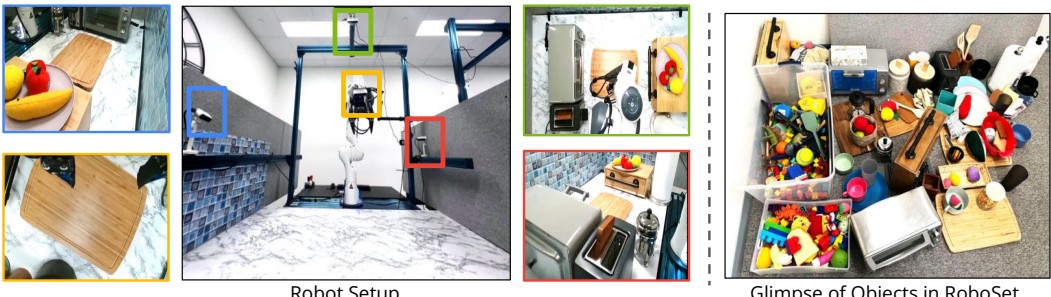

Robot Setup        Glimpse of Objects in RoboSet

Figure 7: A zoomed-out view of the robot environment, showing all four cameras in the scene. *Right:* A glimpse of the diverse objects in `RoboSet`. The objects include articulated objects (drawers, ovens), smaller rigid objects (french press, bowls) and deformable objects (towels, cloth).

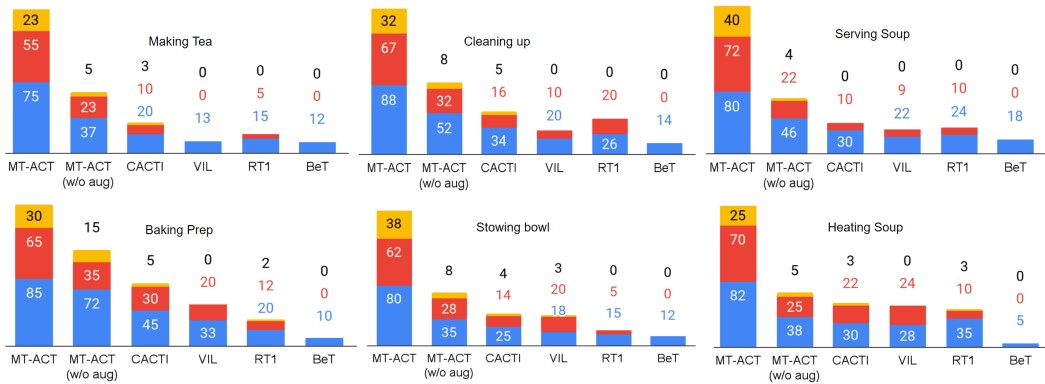

Figure 8: Results for MT-ACT, its ablated variant without semantic augmentations, and baselines, for different activities, with L1, L2, L3 levels of generalization. Each activity consists of 4-5 tasks, and the results average over the tasks in an activity. The results show that semantic augmentations significantly improve performance of MT-ACT over the baselines. In addition, even without augmentations, the MT-ACT policy achieves much higher success rates compared to the baselines.

tion, illustrated visually for a scene in Figure 3: **L1(Effectiveness)**: Generalization of the agent to variations in object positions and orientations, and in lighting conditions. **L2 (Robustness):** New background, different distractor object variations, and unseen distractor objects introduced in the scene. **L3 (Generalization):** New tasks never seen before, including new object-skill combinations. **L4 (Strong Generalization):** New kitchen never seen before (see Figure 9 Left).

# D    Experiments

Through detailed real-world robot manipulation experiments, we evaluate the proposed framework for sample efficiency, and generalization of the agent to diverse scenes. We provide further results (including videos and appendix) on our webpage `https://roboagent-anonymous. github.io/`.

**Baselines.** We compare multiple baselines that use visual policy learning for robotics. *Single Task Agents*: We compare ACT-based policies [8] trained for individual tasks, and evaluated on the respective tasks. These policies don't need to generalize across tasks and scene, and represent an approximate *oracle* performance per task. *Visual Imitation Learning (VIL)*: We compare with regular language-conditioned multi-task visual imitation learning. *CACTI* [29]: This baseline is a prior framework for multi-task learning that also uses some scene augmentations for generalization. *RT1* [5]: We re-implement a baseline RT1-like agent. *BeT* [9]: We modify the Behavior Transformer architecture with language conditioning and train it in a multi-task manner.

**Generalization and Robustness.** Figure 3 (Bottom-Right) shows the results for all methods across multiple levels of generalization (**L1**, **L2**, and **L3**). Recall that these levels of generalization include diverse table backgrounds, distractors (**L2**) and novel skill-object combinations (**L3**). From Figure 3 (Bottom-Right) we see that by virtue of semantic augmentations and action representations, MT-ACT significantly outperforms all the baselines we consider. More interestingly, we see that semantic augmentations have less effect for L1-generalization ($\approx 30\%$ relative), they provide a *much more* significant improvement for both L2-generalization ($\approx 100\%$ relative) and L3-generalization ($\approx 400\%$ relative). Since semantic augmentations affect both scenes (backgrounds and distractor objects) as well as target objects being manipulated they provide useful support for the policy to achieve increasing levels of generalization.

Additionally, in Figure 8 we also separately report generalization results for each activity separately. From Figure 8 we see that our proposed semantic augmentations positively affect each activity's performance. Interestingly, we find that for some of the harder activities (Making-Tea, Stowing-Bowl, Heating Soup) the relative improvement in performance due to semantic augmentations is much larger.

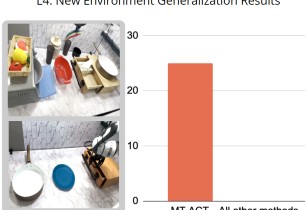

L4: New Environment Generalization Results

Overall, our results show that traditional visual imitation learning (without any augmentations), i.e., VIL and RT1 trained on our relatively small dataset, completely fail for L2 and L3, indicating a lack of generalization to unseen scenarios, due to data paucity. Finally, we also test our policy on a completely new kitchen with novel objects, arrangements, distractors, i.e., L4 generalization. Figure 9 (Left) visualizes this new kitchen environment. We evaluate all methods in this new kitchen for 3 tasks. Figure 9 (Right) shows the results for each method on this new environment. Specifically, we obtain an average success rate of 25% for MT-ACT (and 0 for all baselines). Even MT-ACT without semantic augmentations fails completely on this new environment thus showing the strong generalization ability of our approach.

Figure 9: Only MT-ACT policies perform tasks in a completely new kitchen environment (L4).

### D.1 Ablation studies

**Language conditioning using FiLM.** For language conditioned multi-task policy, as described in section B.3, we use a FiLM based conditioning [63] for the language embedding of task descriptions [68]. We ablate this choice by comparing with simple concatenation-based conditioning. We observe around 10% drop in performance of the version of MT-ACT without FiLM conditioning, across all 4 generalization levels.

**Chunk Size for Action Representations.** We ablate our choice of action chunk size. Figure 10 (Left), shows that a chunk size of 20 performs the best, with a 0-5% drop in performance with chunk size 10. However, a large chunk size 40 performs significantly worse with more than 20% drop in performance indicating the inability of the policy to correct errors as the chunks grow in size.

**Number of augmentations per frame.** Figure 10 (Middle) ablates the number of augmentations per frame, to see if more augmentations help MT-ACT in learning a more performant policy. We see that number of augmentations per frame is strongly correlated with overall performance gains. Thanks to the real-world semantic priors injected via data augmentation, the gains are more notable for L2 and L3 levels where out-of-domain generalization is required.

**Robustness analysis.** We perform robustness analyses of the universal MT-ACT agent, by manually perturbing the scene during evaluation, and also introduce system failures such as blocking camera views. On average, we find that the policy is robust to these strong active variations, and can solve the specified task in about 70% of the 20 evaluations we run for this analysis (videos in the website).

**Plasticity.** We evaluate the feasibility of adding additional capabilities to the universal MT-ACT agent, without requiring significant re-training. We take the trained agent (on 38 tasks) and fine-tune

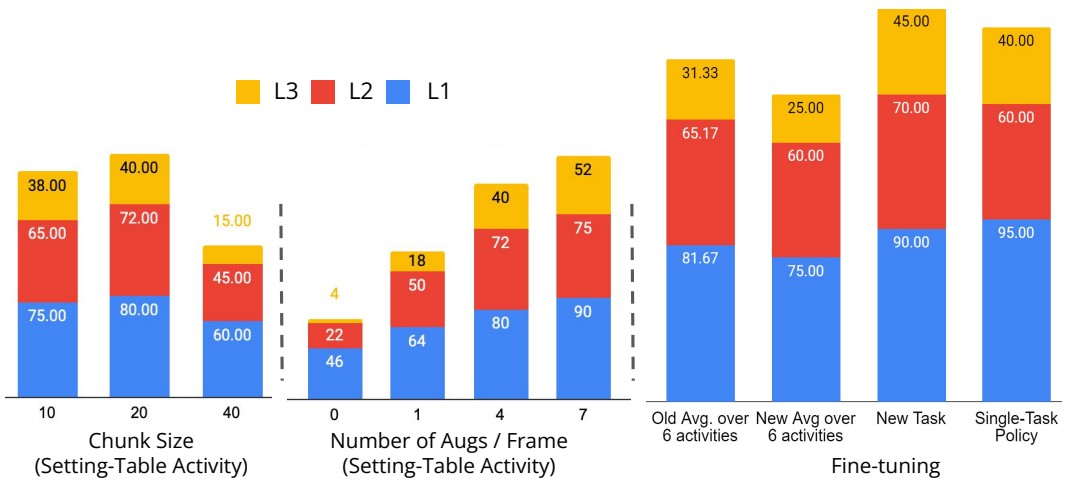

Figure 10: Results for different ablations (see section D.1), showing the benefits of FiLM conditioning, the effect of varying chunk sizes in the predictions, the number of augmentations per frame for multiplying the dataset, and the feasibility of fine-tuning MT-ACT for improved deployment.

on $(1/10)^{\text{th}}$ of the original data combined with data for a new held-out task (placing toast in toaster oven). The new task has 50 trajectories, multiplied with 4 augmentations per frame, for a total of 250 trajectories. Fig. 10 (Right) shows that the fine-tuned agent is able to learn this new task, without significantly deteriorating in performance on the previous 6 activities. Also, it achieves slightly better L2, L3 performance ($\approx 10\%$) than a single-task policy trained only on augmented data of the new task, indicating efficient data re-use.

## D.2  Reproducibility Experiments

To better understand the generalization and plasticity capabilities of *RoboAgent*, we perform a challenging experiment by deploying the trained agent in a completely different location 3000 miles (5000km) away from where data was collected, and observe comparable success rates of 30-60% on new tasks in this setup both for zero-shot deployment and fine-tuning. Video results are in the website https://roboagent-anonymous.github.io/supplementary.html

