# OpenReview forum: "RoboAgent: Generalization and Efficiency in Robot Manipulation via Semantic Augmentations and Action Chunking"
_robot-learning.org/CoRL/2023/Workshop/OOD — OOD Workshop @ CoRL 2023_

### Official Review · Reviewer_4xdg · 2023-10-15
**Addressing OOD generalization**

**Rating:** 8
**Confidence:** 5

**Review:**

The paper proposes a framework called MT-ACT that extends the ACT framework and applies (1) semantic augmentation and (2) language conditioning. The policy is trained with a diverse multi-task dataset. Experiment results demonstrate improvement over baselines in various levels of generalization. Discussions are L3 and L4 generalization (new skill-object pairs and kitchen environments) are particularly fitting for the workshop.

While individual components of the framework are mostly known, the authors nicely combine them together, and each component has been shown effective from previous work and also ablation studies here. Semantic augmentation is a simple yet effective technique for enriching the dataset, and extending the ACT framework to the multi-task setup with language conditioning is intuitive.

I do think the writing can be improved despite the page limit. Adding more details about semantic augmentation in the main text will be welcome.

---

### Official Review · Reviewer_NdLM · 2023-10-16
**Review of RoboAgent**

**Rating:** 10
**Confidence:** 4

**Review:**

**Summary:** This paper proposes a framework for learning generalist robot manipulation skills in a single agent under a limited data budget. Concretely, the approach makes two methodological contributions: 1) automatic techniques for multiplying small datasets with semantic augmentations; 2) an architecture capable of effectively learning from multimodal datasets.

**Relevance:** Automated techniques for data augmentation to support greater OOD generalization is of interest to the manipulation community, and the robotics community more broadly. In the context of this workshop, this paper closely relates to: “Episodic interaction with an environment: Can we develop methods that mitigate or account for shifted conditions that consistently degrade a learning-enabled robot's performance?”

**Novelty:** The automated strategies for semantic augmentation are novel.

**Significance:** The results of this paper are significant. They demonstrate that semantic augmentations and the proposed policy architecture does not only yield performance robustness to new visual features and distractors, but also appears to generalize to entirely new tasks (e.g., manipulating new objects).

**Technical strengths and weaknesses:** The method and evaluation is sound. As a minor suggestion, the reviewer recommends also evaluating the baseline policies with and without semantic augmentations. Such an evaluation would provide insight on the importance of policy architecture in the context of harnessing semantic augmentations.

**Quality / clarity:** The quality and clarity of the paper is high.

**Recommendation:** Accept.

---

### Decision · Program_Chairs · 2023-10-17

**Decision:**

Accept

**Comment:**

We agree with the reviewers’ assessment that this work is technically sound and will contribute to productive, topical discussions at the 2023 Workshop on OOD Generalization in Robotics. In particular, we appreciate that this work explicitly addresses a problem of OOD generalization, and demonstrates quite strong performance along different axes of generalization. We recommend the authors incorporate the reviewers’ feedback into their camera-ready submission to further improve their manuscript.